# HIV-1 Tat interactions with cellular 7SK and viral TAR RNAs identifies dual structural mimicry

Vincent V. Pham[1], Carolina Salguero[1,2], Shamsun Nahar Khan[1,3], Jennifer L. Meagher[4], W. Clay Brown[4], Nicolas Humbert[5], Hugues de Rocquigny[5,6], Janet L. Smith[4,7] & Victoria M. D'Souza[1]

The HIV Tat protein competes with the 7SK:HEXIM interaction to hijack pTEFb from 7SK snRNP and recruit it to the TAR motif on stalled viral transcripts. Here we solve structures of 7SK stemloop-1 and TAR in complex with Tat's RNA binding domain (RBD) to gain insights into this process. We find that 7SK is peppered with arginine sandwich motifs (ASM)—three classical and one with a pseudo configuration. Despite having similar RBDs, the presence of an additional arginine, R52, confers Tat the ability to remodel the pseudo configuration, required for HEXIM binding, into a classical sandwich, thus displacing HEXIM. Tat also uses R52 to remodel the TAR bulge into an ASM whose structure is identical to that of the remodeled ASM in 7SK. Together, our structures reveal a dual structural mimicry wherein viral Tat and TAR have co-opted structural motifs present in cellular HEXIM and 7SK for productive transcription of its genome.

[1] Department of Molecular and Cellular Biology, Harvard University, Cambridge, MA 02138, USA. [2] Vice Presidency of Research, Universidad de los Andes, Bogotá 111711, Colombia. [3] Department of Pharmacy, East West University, Dhaka 1212, Bangladesh. [4] Life Sciences Institute, University of Michigan, 210 Washtenaw Ave, Ann Arbor, MI 48109, USA. [5] Faculté de Pharmacie, Laboratoire de Bioimagerie et Pathologies, UMR 7021 du CNRS, Université de Strasbourg, 74 route du Rhin, 67401 Illkirch, France. [6] Inserm - U1259 MAVIVH. Morphogenèse et Antigénicité du VIH et des Virus des Hépatites, 10 boulevard Tonnelle - BP 3223, 37032 Tours Cedex 1, France. [7] Department of Biological Chemistry, University of Michigan, Ann Arbor, MI 48109, USA. Correspondence and requests for materials should be addressed to V.M.D'S. (email: dsouza@mcb.harvard.edu)

The transition from initiation to elongation during transcription of the integrated HIV genome is critical for expression of the viral mRNA[1–7]. Similar to many cellular genes, RNA polymerase II is able to initiate transcription of the HIV genome but is inhibited soon after by negative elongation factors[8–11]. To relieve this repressed state, stalled polymerases require phosphorylation by the cellular positive elongation factor, pTEFb[2]. However, the availability of pTEFb is tightly regulated and is kept sequestered in an inactive state by the 7SK small nuclear ribonucleoprotein (7SK snRNP) with the help of the HEXIM adapter protein[1,2,4,6,12] (Fig. 1a). To activate elongation, HIV has evolved the viral Tat protein, whose RBD directly interacts with the 7SK snRNA to displace HEXIM and capture pTEFb[13,14]. Tat then transfers pTEFb as part of a larger super elongation complex to the viral genome[15–17]. This handover occurs via the interaction of Tat with the stem loop structure, TAR, which forms at the 5' end of stalled nascent HIV transcripts. The formation of the pTEFb:Tat:TAR ternary complex positions pTEFb in close proximity to the polymerase, thus stimulating elongation[5]. The mechanistic process by which the same Tat RBD is able to engage with both the cellular 7SK and its viral counterpart TAR has thus far remained elusive.

In vivo truncation studies have shown that of the four stem loops in 7SK snRNA, only the first stem-loop is required for Tat interaction, which has been mapped to the upper portion of stem-loop 1 ($G_{24}$ to $C_{87}$; 7SK-SL1$^{apical}$, Fig. 1b) with the remainder of the 7SK RNA being dispensable[14,18,19]. Mutational studies have also shown that the direct Tat:7SK RNA interaction does not require Tat's pTEFb-binding capacity, but is solely dependent on Tat's RNA-binding activity, which is attributed to its RNA binding-domain (RBD, G48-R57)[14]. Furthermore, studies have shown that, like Tat, HEXIM interacts with the same apical portion of the stem-loop 1 in 7SK, with the displacement of HEXIM occurring by direct interaction of Tat with the 7SK snRNA[14,19]. Due to their remarkably similar RBD sequences[13], it has been proposed that Tat has evolved to mimic HEXIM, thus potentially providing a basis for HEXIM displacement.

While there are no structures detailing how Tat RBD binds TAR, on the basis of biophysical studies, it is clear that the hallmark of this interaction is the formation of an arginine sandwich motif (ASM) when an arginine in Tat's RBD intercalates into the bulge region of TAR[20–25]. Specifically, in this motif, nucleotides are arranged in a sandwich-like manner to form stacking interactions with the guanidinium moiety of an arginine: the cap of the motif is formed by a bulge pyrimidine involved in a triple interaction with the stem, while the base of the motif is formed by the nucleotide involved in Watson–Crick pairing immediately preceding the bulge. While structural studies have failed to identify the arginine responsible for the formation of this motif, in vivo experiments have demonstrated arginine 52 (R52) to be the critical residue required for transactivation[20,25–28].

In this study, we provide mechanistic details of how Tat is able to first compete with HEXIM to engage 7SK for pTEFb extraction and subsequently bind TAR to transfer pTEFb.

## Results

**Structure of the free 7SK-SL1$^{apical}$.** We first performed binding studies using Tat RBD (G44-Q60) with 7SK-SL1$^{apical}$ engineered to have a GNRA-type tetraloop in order to prevent aggregation at concentrations needed for solution state biophysical studies. Nuclear Magnetic Resonance (NMR) analysis shows that the various motifs of this construct (described below) are retained despite substitution of the loop (Supplementary Fig. 1a). Additionally, Tat binding studies using isothermal calorimetry

titration (ITC) revealed high-affinity and specific binding traces for both the native loop and GNRA tetraloop constructs ($K_d$ = 51.8 ± 0.7 nM and 32.2 ± 3.5 nM, respectively; Fig. 1c, Supplementary Table 2). To ensure that we were capturing biologically relevant events, we also comparatively studied binding with full-length Tat in the context of the CycT1:Tat:AFF4 complex. This complex binds both the native loop and GNRA tetraloop constructs with comparable affinities ($K_d$ = 55.3 ± 12.5 nM and 44.7 ± 15.4 nM, respectively; Fig. 1c, Supplementary Table 2), indicating that the loop is not critical for RBD binding. Importantly, the affinities of Tat RBD and CycT1:Tat:AFF4 complex are similar, confirming the 7SK-SL1 stem region and Tat RBD to be the primary contributors of the 7SK-snRNA:Tat interaction[13,14,18].

Initial characterization by NMR showed that the folding of the 7SK-SL1$^{apical}$ stem is sensitive to the concentration of salt ions, potentially due to the presence of many bulges interspersed in close proximity within a short stretch of RNA (Fig. 1b). We used Tat RBD binding as a direct readout for correct folding of the RNA. While divalent ions do not have a noticeable effect, low concentrations of monovalent salt (<70 mM NaCl) give rise to non-specific binding of Tat RBD to 7SK-SL1$^{apical}$ (Supplementary Fig. 1b). In stark contrast, tight and specific binding of Tat RBD only occurs when NaCl concentrations are greater than 70 mM. This observation is corroborated by our NMR studies where the RBDs bind in a saturable manner and gives rise to distinct chemical shifts in the slow exchange regime only under ideal monovalent ion concentrations.

Free 7SK-SL1$^{apical}$ is a largely linear molecule with four pyrimidine-rich bulges within one helical turn of the RNA stem (Fig. 1d and Table 1; for details of NMR data used to solve the structure, see Supplementary Discussion). Interestingly, all of these bulges ($C_{75}U_{76}$, $C_{71}U_{72}$, $U_{40}U_{41}$, and $U_{63}$) engage in tertiary interactions that either form or have the potential to form arginine sandwich motifs. The $C_{75}U_{76}$, $C_{71}U_{72}$, and $U_{63}$ bulges form classical arginine sandwich motifs $ASM_1$, $ASM_2$, and $ASM_4$, respectively (Fig. 1e, f). Several unambiguous base-ribose NOE (nuclear Overhauser effect) contacts show that residues $C_{75}$, $C_{71}$, and $U_{63}$ engage in triple interactions with base pairs in the stem ($G_{78}$-$C_{33}$, $G_{74}$-$C_{35}$, and $A_{65}$-$U_{44}$, respectively) to form the caps of the sandwiches. On the other hand, residues $G_{74}$, $G_{70}$, and $C_{62}$ of the Watson–Crick pairs that precede these caps form the bases of the sandwiches (Fig. 1d–f and Supplementary Figs 2–4, and Supplementary Discussion).

In stark contrast, the $U_{40}U_{41}$ bulge forms a pseudo-arginine sandwich motif (pseudo-$ASM_3$) with a unique structural architecture wherein the sandwich cap is preformed but the sandwich base is sequestered. Specifically, $U_{40}$ forms a conventional cap by engaging with $A_{43}$-$U_{66}$ to form a triple interaction. However, intense NOE connectivities from both the H1' and H8 protons of the preceding $A_{39}$ to the H3 proton of $U_{68}$ show that $A_{39}$ is sequestered in a reverse Hoogsteen interaction (Supplementary Fig. 2). This orients the $A_{39}$ purine ring towards the minor groove and precludes it from forming the conventional base of an ASM (Fig. 1e). Out of the four motifs present in free 7SK-SL1$^{apical}$, only the pseudo-$ASM_3$ has a unique architecture. As this is the only ASM where the sandwich base is not part of a G–C Watson–Crick pair, we mutated the reverse Hoogsteen $A_{39}$o$U_{68}$ pair to a $G_{39}$-$C_{68}$ pair ($A_{39}$G, $U_{68}$C). This change allowed the motif to take on a completely preformed characteristic like the other ASMs, confirming that sequestration of the sandwich base in pseudo-$ASM_3$ is caused by the presence of an AoU pair preceding the bulge (Supplementary Fig. 5a).

The orientation of the individual motifs gives rise to two structural entities based on their relative proximities. First, $ASM_1$

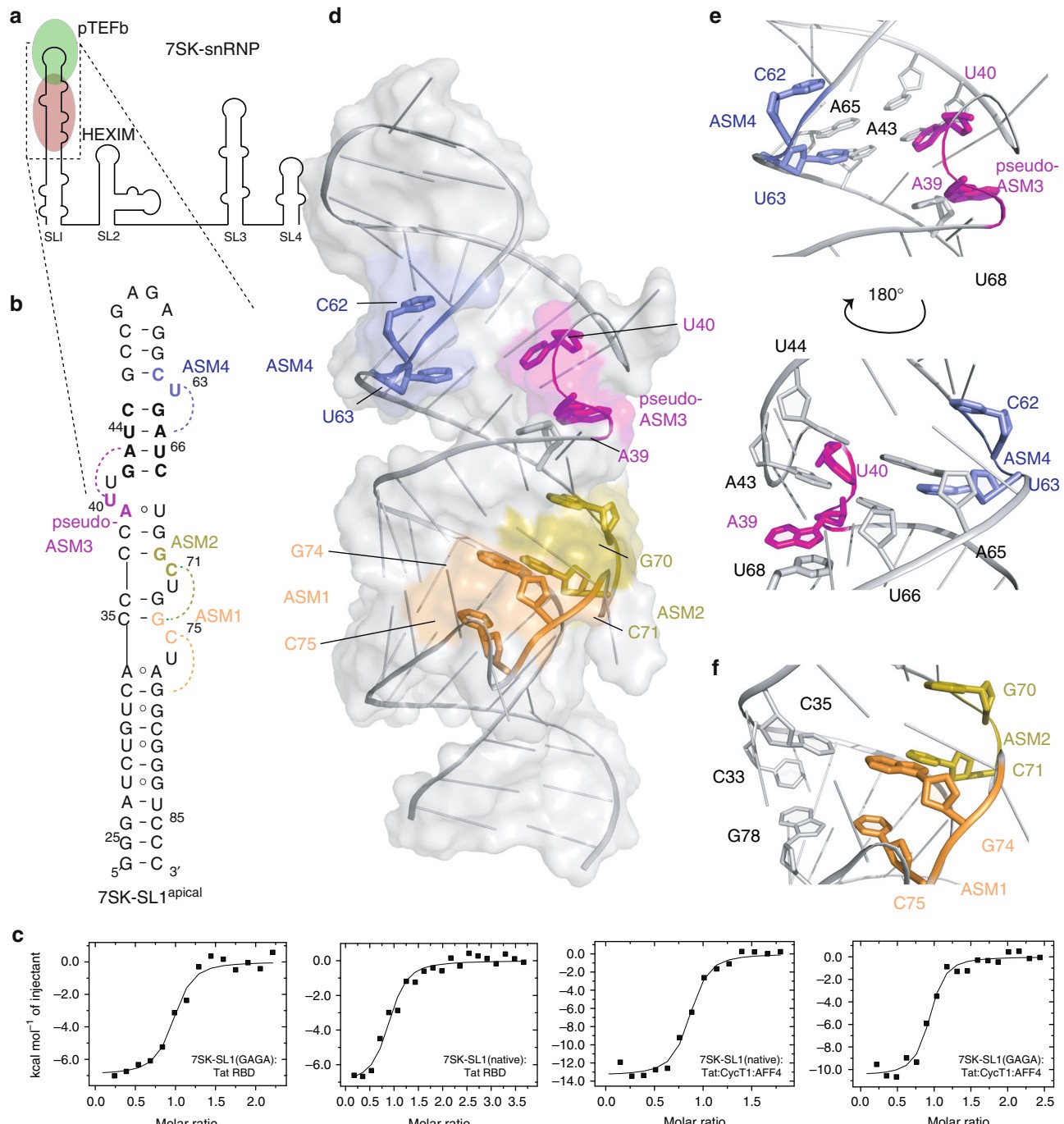

**Fig. 1** Structure and characterization of free 7SK-SL1$^{apical}$. **a** Cartoon representation of pTEFb and HEXIM binding to 7SK-SL1 in 7SK snRNP. **b** Secondary structure of 7SK-SL1$^{apical}$ with the base and cap residues forming ASM$_1$, ASM$_2$, pseudo-ASM$_3$, and ASM$_4$ colored in orange, olive, magenta, and blue, respectively. The dashed arcs represent the triple-base interactions from the bulge to the stem, giving rise to the roofs of the sandwiches. Nucleotides in bolded black represent the four palindromic base pairs involved in pseudo-ASM$_3$ and ASM$_4$ formation. **c** Representative ITC data for Tat RBD and Tat: CycT1:AFF4 binding to 7SK-SL1$^{apical}$ with a GNRA or native loop, which show Tat RBD-mediated interactions are preserved in the larger complex. The continuous lines represent the fit for a one-site binding model. **d** NMR structure of the free 7SK-SL1$^{apical}$ RNA with color schemes matching the secondary structure in **b**. **e** 180° rotated views of the apical, near-symmetric pseudo-ASM$_3$ and ASM$_4$ shows residues U$_{40}$ and U$_{63}$ forming triple bases with consecutive stem base pairs and A$_{39}$ sequestered in a reverse Hoogsteen base pair. **f** View of the tandem ASM$_1$ and ASM$_2$ shows that G$_{74}$ is involved in the formation of both sandwiching motifs

and ASM$_2$ are oriented in tandem on the same strand in the lower part of the stem loop. Both of these ASMs share residue G$_{74}$ to form overlapping sandwiches. Specifically, the C$_{75}$ cap of ASM$_1$ is engaged in a typical C$_{75}$:G$_{78}$-C$_{33}$ triple interaction, and residue

G$_{74}$ forms the base of the sandwich; however, G$_{74}$ also simultaneously engages the protonated C$_{71}$ cap of ASM$_2$ in a C$_{71}$$^+$:G$_{74}$-C$_{35}$ triple-base interaction (Fig. 1f and Supplementary Figs. 2, 3). Second, the pseudo-ASM$_3$ and ASM$_4$ are arranged on

**Table 1 NMR statistics and restraints for 7SK-SL1$^{apical}$, 7SK-SL1$^{apical}$:Tat RBD, and Tat RBD:TAR**

| | 7SK-SL1 | | 7SK-SL1:Tat RBD (CYANA) | | | 7SK-SL1:Tat RBD (AMBER) | | |
|---|---|---|---|---|---|---|---|---|
| | (CYANA) | (AMBER) | 7SK-SL1 | Tat RBD | Complex | 7SK-SL1 | Tat RBD | Complex |
| **NMR derived restraints** | | | | | | | | |
| Distance restraints | | | | | | | | |
| Total NOE | 727 | 556 | 806 | 33 | 807 | 632 | 33 | 633 |
| Intra-residue | 421 | 251 | 430 | – | 430 | 260 | – | 260 |
| Inter-residue | 175 | 175 | 213 | 1 | 214 | 209 | 1 | 210 |
| H-bond restraints | 131 | 130 | 163 | 32 | 163 | 163 | 32 | 163 |
| Torsion angle restraints | 696 | – | 692 | 314 | 1006 | – | – | – |
| Protein | – | – | – | 314 | 314 | – | – | – |
| Nucleic acid | 696 | – | 692 | – | 692 | – | – | – |
| **Structure statistics** | | | | | | | | |
| Violations | | | | | | | | |
| Mean AMBER energy (kcal mol$^{-1}$) | – | −13,661.40 | – | | | −15,206.87 | | |
| Mean constraint energy (kcal mol$^{-1}$) | – | 18.95 | – | | | 36.96 | | |
| Distance violations (>0.5 Å) | – | 0 | – | | | 0 | | |
| Dihedral angle violations (>5 °C) | – | 0 | – | | | 0 | | |
| Average pairwise r.m.s. deviation (Å)$^a$ | | | | | | | | |
| Tat RBD (51–57) | – | – | – | | | 0.57 ± 0.25 | | |
| 7SK-SL1$^{apical}$ (16–52,60–87) | – | 0.47 ± 0.10 | – | | | 0.67 ± 0.14 | | |
| 7SK-SL1$^{apical}$:Tat RBD | – | – | – | | | 0.79 ± 0.25 | | |

| | TAR:Tat RBD (CYANA) | | | TAR:Tat RBD (AMBER) | | |
|---|---|---|---|---|---|---|
| | TAR | Tat RBD | Complex | TAR | Tat RBD | Complex |
| **NMR derived restraints** | | | | | | |
| Distance restraints | | | | | | |
| Total NOE | 425 | 30 | 425 | 335 | 30 | 335 |
| Intra-residue | 234 | – | 234 | 144 | – | 144 |
| Inter-residue | 96 | 5 | 96 | 96 | 5 | 96 |
| H-bond restraints | 95 | 25 | 95 | 95 | 25 | 95 |
| Torsion angle restraints | 373 | 315 | 688 | – | – | – |
| Protein | – | 315 | 315 | – | – | – |
| Nucleic acid | 373 | – | 373 | – | – | – |
| RDC Restraints | – | – | 11 | – | – | 11 |
| **Structure statistics** | | | | | | |
| Violations | | | | | | |
| Mean AMBER energy (kcal mol$^{-1}$) | – | | | -8675.25 | | |
| Mean constraint energy (kcal mol$^{-1}$) | – | | | 36.26 | | |
| Distance violations (>0.5 Å) | – | | | 0 | | |
| Dihedral angle violations (>5 °C) | – | | | 0 | | |
| RDC violations (>0 Hz) | | | | 0 | | |
| Average pairwise r.m.s. deviation (Å)$^a$ | | | | | | |
| Tat RBD (49–54) | – | 0.91 ± 0.30 | | | | |
| TAR RNA (16–46) | – | 0.54 ± 0.25 | | | | |
| TAR:Tat RBD | – | 0.84 ± 0.19 | | | | |

$^a$Pairwise r.m.s. deviation was calculated among 10 refined structures

opposite strands with a near-symmetrical architecture (Fig. 1e). The sandwich caps of the pseudo-ASM$_3$ (U$_{40}$) and ASM$_4$ (U$_{63}$) form symmetrical triple interactions with palindromic base pairs in the stem (U$_{40}$:A$_{43}$-U$_{66}$, and U$_{63}$:A$_{65}$-U$_{44}$, respectively). Hence, despite being four base pairs apart in the secondary structure, this architecture arranges them as consecutive triple bases. Indeed, this close spatial proximity of the cap residues U$_{40}$ and U$_{63}$ is evidenced by connectivities between their H3 protons (Supplementary Fig. 2a, b). On the other hand, the arrangement of the sandwich bases prevents the formation of a truly symmetrical unit due to the differential geometries of the reverse Hoogsteen and Watson-Crick pairing in pseudo-ASM$_3$ and ASM$_4$, respectively (Fig. 1e). Overall, these distinct structural units present 7SK with two different platforms for arginine interaction: a pair of tandem motifs provides two

adjacent preformed cavities, while the other spatially opposed, partially-symmetric pair provides one preformed cavity and one sequestered motif.

**Engagement of 7SK-SL1$^{apical}$ by Tat's RBD.** To investigate how the full-length Tat RBD utilizes the various arginine-sandwich motifs in 7SK, we performed structural analyses by SAXS and NMR. The reconstructed ab initio SAXS envelope shows that there are no major rearrangements in the global architecture of the 7SK-SL1$^{apical}$ upon protein binding (Fig. 2a). Indeed, while we observe numerous intermolecular NOEs from all motifs, no major changes in NOE connectivities that define the preformed sandwiches themselves were observed, proving that the overall structural integrity is maintained (Fig. 2b and Table 1,

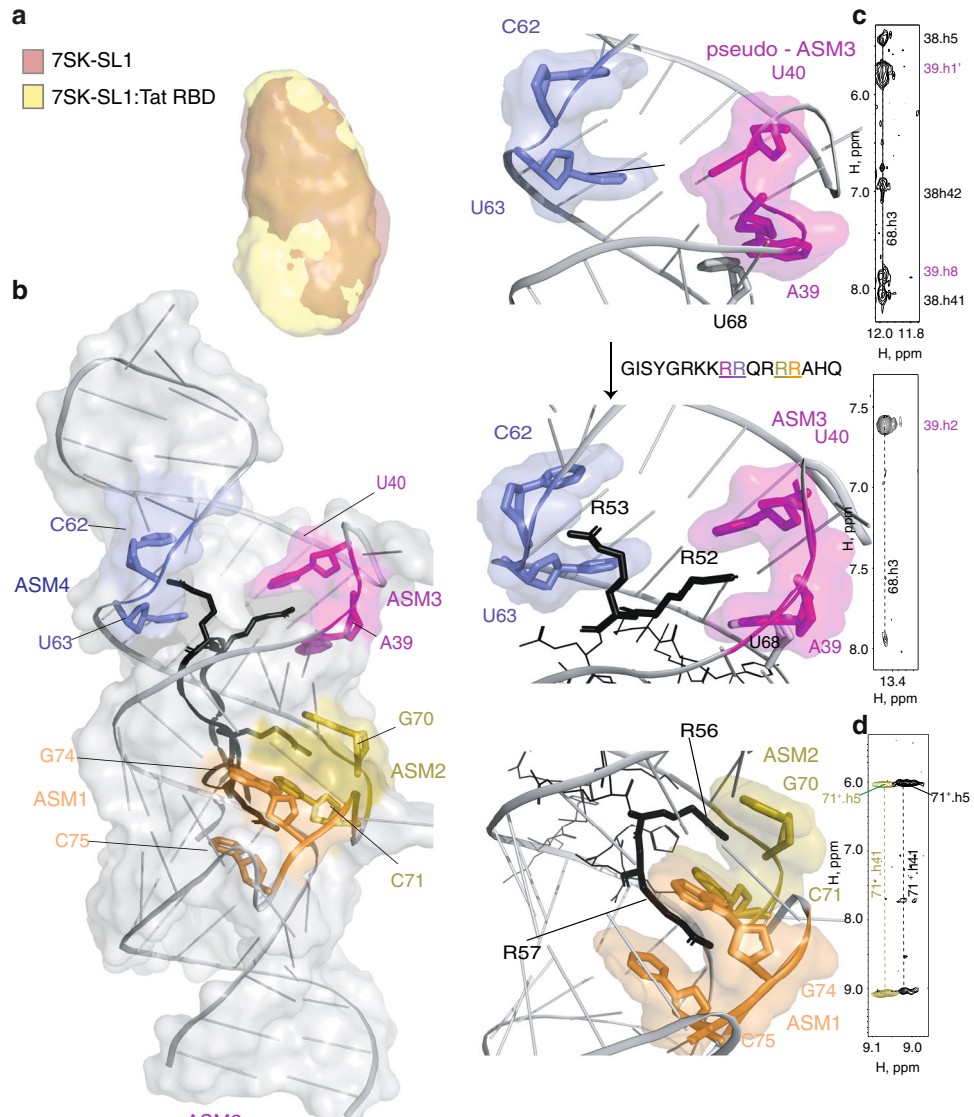

**Fig. 2** Structure and characterization of Tat RBD bound to 7SK-SL1$^{apical}$. **a** Overlay of reconstructed ab initio SAXS envelope of free 7SK-SL1$^{apical}$ (coral) and the 7SK-SL1$^{apical}$:Tat RBD complex (yellow) demonstrating the lack of global rearrangement of the RNA. **b** NMR structure of 7SK-SL1$^{apical}$ bound to Tat RBD (black), showing multiple arginines intercalating into the ASMs. Arginine residues in the Tat RBD primary sequence are color-coded to match the respective ASMs that they intercalate into. Top and middle insets: R52 induces the formation of ASM$_3$ from the pseudo-ASM$_3$ configuration, while R53 interacts with the preformed ASM$_4$. Bottom inset: interaction of the tandem ASM$_1$ and ASM$_2$ with R57 and R56, respectively. **c** $^1$H–$^1$H two- dimensional NOESY showing transition of A$_{39}$-U$_{68}$ from a reverse Hoogsteen interaction to a Watson-Crick interaction by the appearance of the characteristic U$_{68}$ imino to A$_{39}$ H2 upon titration of Tat. **d** Spectral overlay showing that ASM$_2$ remains intact after Tat intercalation as evidenced by the maintenance of the downfield shifted amino of the protonated cap C$_{71}$+. Black, free 7SK-SL1$^{apical}$ and olive, after titration with 1.0 equivalent of Tat RBD

See Supplementary Discussion). Most perturbations occur in pseudo-ASM$_3$. In particular, the A$_{39}$ H2 proton experiences a dramatic upfield chemical shift from 6.77 to 6.49 ppm and gives rise to an intense NOE connectivity to the U$_{68}$ H3 imino proton characteristic of a Watson-Crick base pair formation between the two residues (Fig. 2c). Thus, upon Tat RBD binding, the A$_{39}$oU$_{68}$ reverse Hoogsteen becomes a Watson–Crick base pair, converting the pseudo into a classical arginine sandwich motif (ASM$_3$) to give rise to a truly symmetrical arrangement of ASM$_3$ and ASM$_4$ that results in four fully-formed ASMs (Fig. 2c).

The structure shows that residues K51-R57 of Tat RBD engage 7SK-SL1$^{apical}$. Although the arginine sandwich motifs are dispersed throughout an entire helical turn of the RNA, this eight-amino acid stretch is able to engage all four ASMs,

potentially due to the multiple points of arginine intercalations into the major groove from the start to the end of the helical turn (Supplementary Discussion). To position the various arginines into the correct arginine sandwich motifs, we used Tat RBD with various combinations of specifically labeled amino acids (Supplementary Fig. 5b). An unambiguous set of intermolecular NOEs places R57, R56, R52, and R53 into ASM$_1$ through ASM$_4$, respectively (Supplementary Fig. 6). The arginine side chains are precisely anchored by intermolecular NOEs between the arginine side chain protons with the aromatic H5/H6 protons of the sandwich caps. For example, R53 and R52 Hβ and Hγ protons gave strong NOE connectivities to the U$_{63}$ and U$_{40}$ H5 protons, respectively (Fig. 3a and Supplementary Fig. 6a, b). We also observe guanidinium nitrogen moieties from all labeled arginines

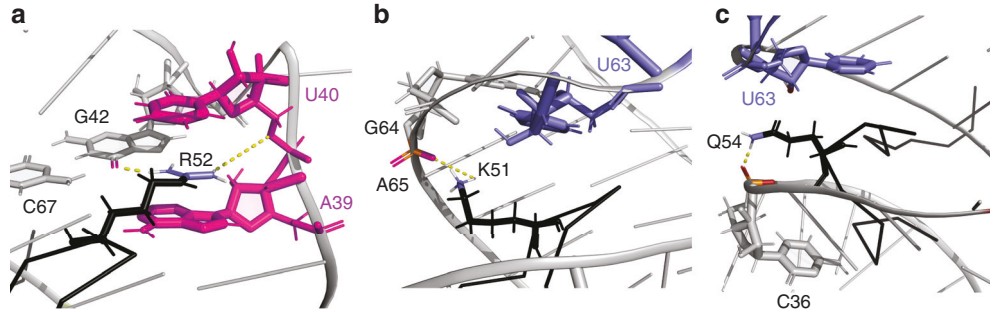

**Fig. 3** Intermolecular interactions between Tat RBD and 7SK-SL1$^{apical}$. Close-up view of **a** the ASM$_3$ binding pocket showing residues G$_{42}$ (gray), A$_{39}$ and U$_{40}$ (magenta), and R52 (black). The dotted lines indicate hydrogen bond interactions from the two N groups of arginine 52 to the O6 of G$_{42}$ and the O5′ of U$_{40}$; **b** the K51 interaction as Tat RBD exits the major groove. The dotted lines show the close distance of the side chain amino group of K51 to the phosphate of A$_{65}$, showing the potential to form a hydrogen bond; **c** the interaction of the spacer residue Q54 within the major groove that places it under the U$_{63}$ cap residue

after complex formation, indicating the slow-exchange of these Hη protons due to their involvement in hydrogen-bonding interactions within the sandwiches (Supplementary Fig. 5b).

Both the tandem and the symmetric cavities accommodate consecutive arginines. Since the tandem ASM$_1$ and ASM$_2$ stack on the same strand, it allows the C-terminal arginines R57 and R56 to dock into these motifs in a ladder-like configuration (Fig. 2b, Supplementary Fig. 6c). This occurs without any significant changes in the organization of these ASMs (Fig. 2d). Tat RBD then continues its course through the major groove and docks the N-terminal arginine R52 into ASM$_3$ (Fig. 2b). However, the preceding arginine residue R53 flips upward to interact with ASM$_4$, causing the chain to reverse its direction and exit the RNA through the same groove as the C- terminus. This inverse intercalation of R52 and R53 into the symmetrical ASM$_3$ and ASM$_4$ creates a distinct fork-like arrangement of the two arginines (Fig. 2b). The reversal of the chain also leads to the formation of a short β-hairpin encompassing residues K51 to Q54. Intermolecular connectivities between K51 Hε and the H8 of A$_{65}$ support this reversal and places the side-chain amino group of K51 within hydrogen bonding distance of the A$_{65}$ phosphate backbone (Fig. 3b and Supplementary Fig. 6d). Q54 and R55 serve as spacer residues that bridge the distance between the symmetrical and tandem motifs, allowing the N-terminal and C-terminal arginines to intercalate into these motifs, respectively. This is evidenced by connectivities from R55 Hδ to H5/H6 of C$_{35}$ and Q54 Hβ to the H2′ of the U$_{63}$ roof, placing Q54 directly under ASM$_4$ (Fig. 3c and Supplementary Fig. 6a). Taken together, our structures show how the highly conserved arginine-rich sequence intimately docks deep into the major groove using all four arginine sandwiches as anchoring points.

To understand the contributions of each ASM in 7SK-SL1$^{apical}$ for Tat binding, we designed constructs to individually disrupt the formation of each of the ASMs (ASM$_1$$^{U76A}$, ASM$_2$$^{U72A}$, ASM$_3$$^{U40A}$, and ASM$_4$$^{ΔU63}$). These mutations not only successfully abrogate the intended ASM, but also do not significantly affect the integrity of the other ASMs (Supplementary Figs. 7, 8a). Binding analysis via ITC shows that while TatRBD binds ASM$_1$$^{U76A}$, ASM$_2$$^{U72A}$, and ASM$_4$$^{ΔU63}$ with a ten-fold, three-fold and thirteen-fold decreased affinity ($K_d = 308.7 ± 30.2$ nM, $86.8 ± 34.9$ nM, and $428.3 ± 221.4$ nM), respectively (Supplementary Fig. 8b, Supplementary Table 2), binding of Tat-RBD to the ASM$_3$$^{U40A}$ construct is completely abolished (Supplementary Fig. 8c). These data are consistent with in vivo work, which show that deletions in the ASM$_1$, ASM$_2$ region, and a deletion of U$_{63}$ in ASM$_4$ are tolerated, but a single mutation at the U$_{40}$ position is detrimental for Tat binding[14].

**Mechanism of HEXIM displacement by Tat**. To gain a mechanistic understanding of how Tat displaces HEXIM, we performed biochemical and structural studies with HEXIM RBD (K149-R156). The binding affinity of HEXIM for 7SK-SL1$^{apical}$ is two-fold weaker in comparison to Tat ($K_d = 66.6 ± 9.0$ nM and $32.2 ± 3.5$ nM, respectively; Fig. 1c, Fig. 4a, Supplementary Table 2), although both have similar modes of binding in that they are both enthalpically and entropically favorable ($ΔH = −5.9 ± 1.5$ and $−7.9 ± 1.7$ kcal mol$^{-1}$ and $−TΔS = −4.33 ± 1.4$ and $−1.95 ± 1.5$ kcal mol$^{−1}$, respectively). Binding experiments of the native 7SK-SL1$^{apical}$ loop to HEXIM and Tat also show that the difference in binding affinities is maintained in the native loop context ($K_d = 80.0 ± 1.8$ nM and $55.3 ± 12.5$ nM, respectively; Figs. 1c, 4a, Supplementary Table 2).

Next, we performed competition experiments by NMR. We began by titrating HEXIM RBD into 7SK-SL1$^{apical}$ in the NMR to understand HEXIM's mode of binding. While HEXIM's RBD is able to engage the preformed motifs, no signature A$_{39}$ H2 chemical shift indicative of ASM$_3$ formation is observed, even upon adding five-fold excess of HEXIM RBD (Fig. 4b). We then performed a direct competition assay by titrating Tat RBD into this HEXIM RBD:7SK-SL1$^{apical}$ complex in order to visualize any structural changes that occur. Addition of Tat's RBD causes the emergence of the upfield-shifted A$_{39}$ H2 proton at 6.49 ppm, showing that Tat RBD is able to outcompete HEXIM and bind 7SK-SL1$^{apical}$ (Fig. 4b). On the other hand, titration of HEXIM RBD into a 7SK-SL1$^{apical}$:Tat complex does not result in the reversal of the A$_{39}$ H2 proton chemical shift, indicating that HEXIM is unable to effectively compete with Tat RBD (Fig. 4b).

Since the transition from pseudo-ASM$_3$ to ASM$_3$ accompanies HEXIM displacement and represents the only major change in 7SK induced by R52 upon Tat encounter, we wanted to understand the interplay between R52 and ASM$_3$, which allows for both HEXIM displacement and Tat binding to 7SK. In vivo data shows that like Tat, HEXIM binding is critically dependent on the pseudo-ASM$_3$ region as even minor changes to residues in this region completely abolishes HEXIM recognition[13,14,19]. Indeed, while titration of HEXIM RBD into 7SK-SL1$^{apical}$ gives rise to high affinity binding, titration into the ASM$_3$$^{U40A}$ construct, like Tat, leads to no heats of binding (Fig. 4a, c). Since the same RNA region is important for both Tat and HEXIM binding and this bulge can adopt both a pseudo- and a classical ASM configuration, we then wanted to test if there are any predetermined requirements that specify either Tat or HEXIM RBD binding. To accomplish this, we used the A$_{39}$G, U$_{68}$C construct where the pseudo-ASM$_3$ was engineered to force the formation of a fully-formed, classical ASM$_3$. While Tat RBD

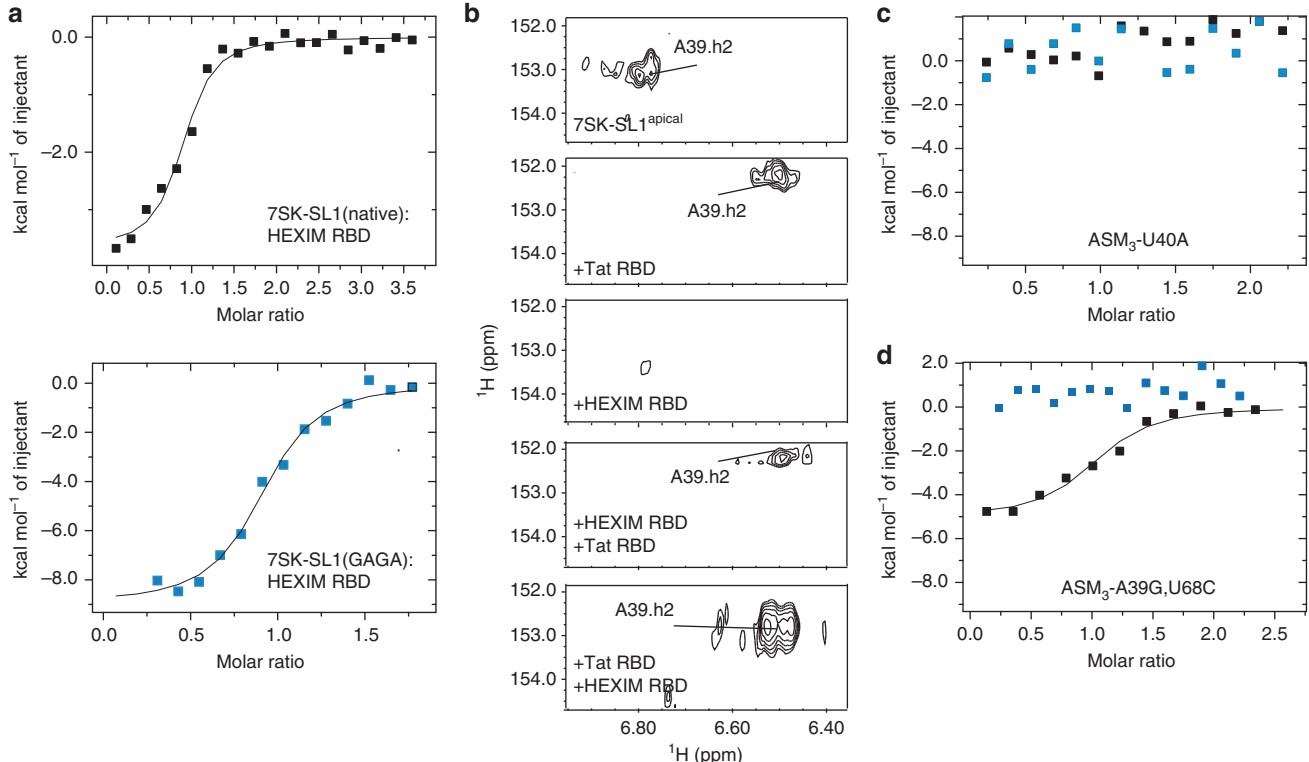

**Fig. 4** The role of R52 in HEXIM displacement. **a** Representative ITC data for HEXIM RBD binding to both 7SK-SL1$^{apical}$ both with the native (top) and GAGA tetraloop (bottom). A continuous line represents the fit for one-site binding model. **b** HMQC spectra of free 7SK-SL1$^{apical}$ (top panel). Titration of 1 equivalent of Tat RBD leads a characteristic upfield shift in the A$_{39}$ H2 proton indicative of the A$_{39}$-U$_{68}$ Watson-Crick base-pair formation (second panel). On the other hand, titration of five equivalents of HEXIM RBD did not lead to such a shift (third panel). Titration of 1 equivalent of Tat RBD into this HEXIM:7SK complex led to the emergence of the upfield A$_{39}$ chemical shift characteristic of that of Tat:7SK-SL1$^{apical}$, indicating HEXIM displacement (fourth panel). On the other hand, titration of 1 equivalent of HEXIM RBD into a Tat:7SK-SL1$^{apical}$ complex with 2 equivalents of Tat RBD did not lead to the loss of the upfield-shifted A$_{39}$ H2 proton shift, indicting lack of Tat displacement by HEXIM (bottom panel). **c** Representative ITC data for both HEXIM RBD (blue) and Tat RBD (black) binding to the U$_{40}$A mutant, which disrupts the formation of pseudo-ASM$_3$. Both constructs gave no heats of binding. **d** Representative ITC data for titration of both HEXIM (blue) and Tat (black) RBDs into the A$_{39}$G, U$_{68}$C construct that causes a preformed ASM$_3$. While Tat RBD was able to engage this motif, HEXIM RBD gave no heats of binding

binds this construct, albeit with a five-fold decrease in affinity compared to the wild-type ($K_d = 140.5 \pm 30.9$ nM), HEXIM RBD binding was abolished (Fig. 4d, Supplementary Table 2), showing that classical sandwich formation and HEXIM binding are mutually exclusive. Furthermore, R52K is also unable to bind the preformed ASM$_3$ construct with specificity, showing that a lysine at this position prohibits both the R52K mutant Tat and HEXIM from appropriately engaging a classical ASM$_3$ (Supplementary Fig. 8d,e). Taken together, these data indicate that the ability of Tat to switch the pseudo-ASM$_3$ into a classical sandwich due to a single additional arginine provides Tat a competitive advantage for displacing HEXIM.

To understand the importance of arginine intercalations for HEXIM displacement, we mutated individual intercalating arginines to alanines. While the data for R53A binding was uninterpretable due to severe line-broadening of NMR signals, studies with the R57A, R56A, and R52A mutant constructs show that interactions of individual arginines occur independently of the neighboring ASM (Supplementary Fig. 8f). These constructs are also able to compete with HEXIM for binding to 7SK-SL1$^{apical}$, albeit with reduced affinity. Concomitant with binding, both R56A and R57A are able to induce the transition of ASM$_3$ from pseudo to classical while R52A is unable to do so (Supplementary Fig. 9). Our structures show R52 to be responsible for the classical sandwich formation, and since it also represents the only conserved substitution from K151 in HEXIM (KR$_{52}$RQRRR in

Tat vs. KK$_{151}$KHRRR in HEXIM)[13], we studied the reciprocal Tat R52K and HEXIM K151R mutations. While these 7SK-SL1$^{apical}$ complexes are able to engage the tandem ASMs (ASM$_1$ and ASM$_2$), their interactions with the symmetrical motifs (ASM$_3$ and ASM$_4$) were dynamic (Supplementary Fig. 9). The HEXIM K151R construct is able to remodel the pseudo configuration and is not completely displaced from 7SK-SL1$^{apical}$ upon encountering native Tat RBD (Supplementary Fig. 9). Furthermore, the R52K complex reveals that unlike the R52A mutation, R53 can compensate for the remodeling of ASM$_3$ in the absence of R52, indicating that the local environment of the RBD encountering the pseudo-ASM$_3$ dictates the ability to adapt and remodel this motif (Supplementary Fig. 8f). This compensatory role in inducing the switch leads to complete displacement of HEXIM upon titration of R52K into a HEXIM:7SK-SL1$^{apical}$ complex (Supplementary Fig. 9). These studies show the advantage of having multiple intercalation points, which allows for plasticity both in the mode of binding to 7SK-SL1$^{apical}$ and for HEXIM displacement.

**Engagement of TAR by Tat's RBD.** Similar to 7SK-SL1$^{apical}$, our ITC experiments show high affinity binding of both the Tat RBD and the CycT1:Tat:AFF4 complex to TAR ($K_d = 22.5 \pm 15.2$ nM and $77.7 \pm 62.7$ nM, respectively), confirming previous studies that the Tat RBD significantly contributes to the interaction with TAR RNA (Fig. 5a, b, Supplementary Table 2)[23,29,30]. We also confirmed that the CycT1:Tat:AFF4 complex is able to engage the

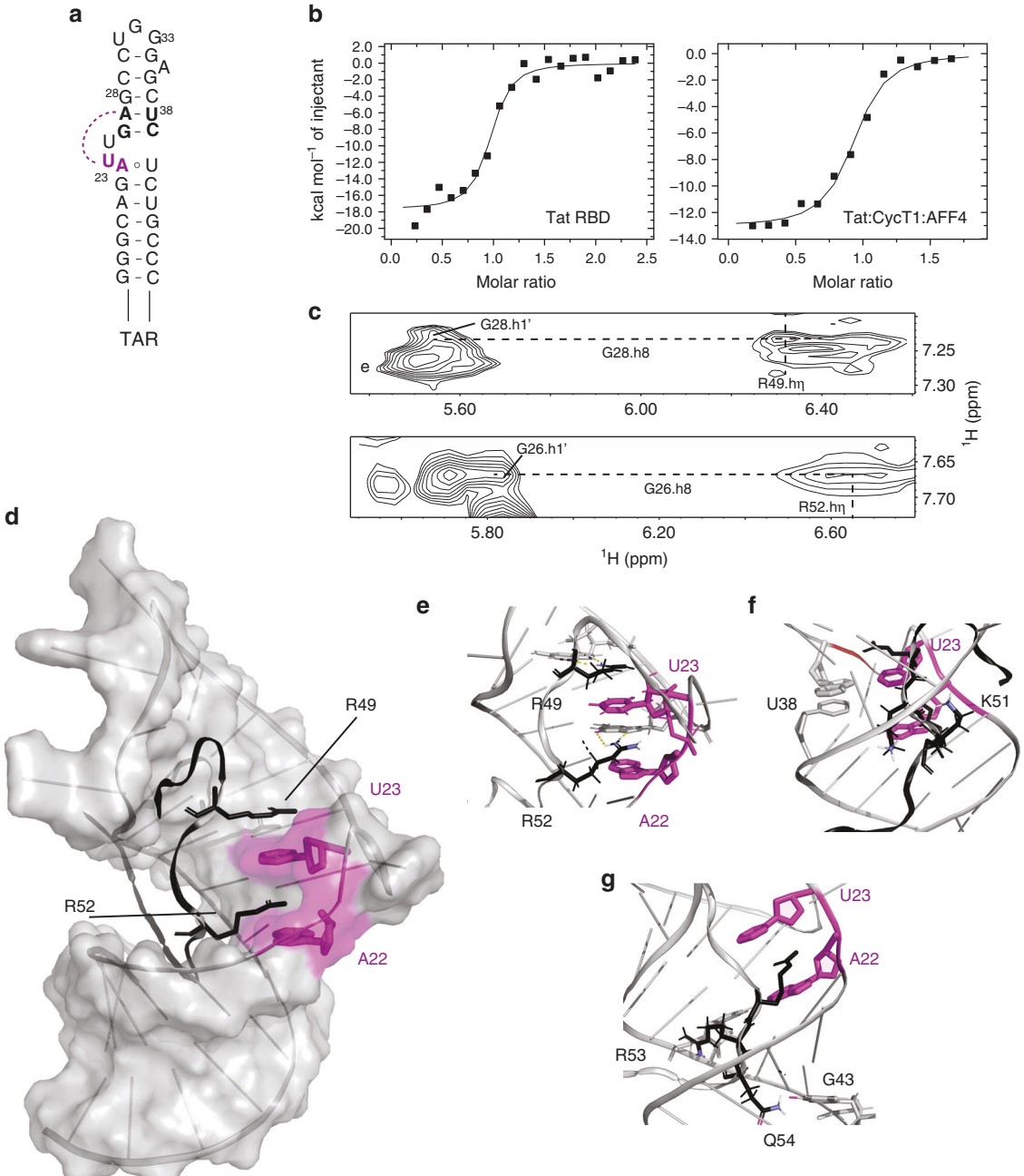

**Fig. 5** Structure and characterization Tat RBD in complex with TAR. **a** Secondary structure of TAR where the dashed arcs represent the triple-base interactions between $U_{23}$ from the bulge to the $A_{27}$-$U_{38}$ base pair in the stem, giving rise to the roof of the sandwich. **b** Representative ITC data of Tat RBD (left) and the Tat:CycT1:AFF4 (right) titration into TAR RNA, respectively. Continuous lines represent the fit for a one-site binding model. **c** $^1$H–$^1$H two-dimensional NOESY showing the NOEs between the R49 and R52 hη guanidinium protons and $G_{28}$ and $G_{26}$ H8 protons, respectively, placing these residues in an arginine fork interaction. **d** NMR structure of TAR bound to Tat RBD (black) showing R49 and R52 sandwiching the $U_{23}$ cap of the ASM sandwich. **e** The ASM binding pocket with the $A_{22}$ base and the $U_{23}$ cap (magenta), $G_{26}$ and $G_{28}$ (gray), and R49 and R52 (black). The dotted lines indicate hydrogen bond interactions from the NH group of arginine 49 and 52 to the O6 and N7 of $G_{28}$ and $G_{26}$, respectively. **f** Positions of K50 and K51 (black) in the major groove. **g** The R53 and Q54 interaction with U40 and G43, respectively. Close proximity of the R53δ to the $C_{39}$, $U_{40}$, and $C_{41}$ H5 protons evidenced by the data and a dotted line represents the potential hydrogen bonding interaction between a guanidinium proton of R53 to the $U_{40}$ phosphate. The hydrogen bond interaction between the Q54 hε proton and the $G_{43}$ O6 represents the C-terminal exit of Tat RBD from TAR

TAR loop in the same manner as previous established (Supplementary Fig. 10a)[31]. To make structure-based comparisons between 7SK-SL1[apical] and TAR, we solved the structure of Tat RBD:TAR complex under the same salt conditions used to solve the Tat RBD: 7SK-SL1[apical] complex.

As predicted from previous work, titration of Tat RBD into TAR results in the formation of an arginine sandwich motif[22,23,32]: $U_{23}$ of the bulge makes a triple-base interaction with $A_{27}$-$U_{38}$ base pair in the stem to form the cap of the sandwich while the preceding $A_{22}$ forms the base of the sandwich

(Supplementary Fig. 10b). It is interesting to note that the secondary structures of the TAR bulge and 7SK-SL1$^{apical}$ ASM$_3$ are identical. While Tat RBD binding eventually does give rise to identical tertiary arginine sandwich structures, the free structures are significantly different. Unlike the pseudo-ASM$_3$ configuration in 7SK where the base triple is preformed, the TAR bulge is flexible and adopts multiple conformations. We rationalized that this stark difference must arise due to effects from neighboring motifs: whereas in 7SK-SL1$^{apical}$ the pseudo-ASM$_3$ triple base is in close proximity to a structured ASM$_4$, in TAR, the preformed triple base would be in close proximity to the dynamic apical hexaloop[33]. Indeed, substitution of the TAR hexaloop with a GNRA motif allows the bulge to take on the pseudo-ASM configuration, indicating that the apical loop has a destabilizing effect on the bulge (Supplementary Fig. 10c).

Upon Tat RBD binding, this apical loop is stabilized by the formation of a C$_{30}$-G$_{34}$ base-pair, thus giving rise to the predicted pseudo-triloop configuration (Supplementary Fig. 10b)[34–36]. G$_{26}$ though U$_{31}$ show NOEs representative of classical stacking interactions, after which there is a strand reversal with G$_{32}$ and G$_{33}$ oriented towards the minor groove face of the molecule. The strand then continues with G$_{34}$ stacking onto G$_{36}$ and with A$_{35}$ flipped out of the stem, positioning the base towards the minor groove as evidenced by only a weak A$_{35}$ H2 connectivity to the G$_{36}$ H1' (Supplementary Fig. 11).

The structure of the TAR:Tat RBD complex shows that Tat RBD enters the major groove of TAR near the stem-pentaloop junction, as evidenced by a network of NOEs from the G$_{28}$-C$_{37}$ basepair and the closing pentaloop base, G$_{34}$, to the N-terminal R49. Briefly, the R49 Hβ and Hγ protons are in close proximity to G$_{34}$ H8, whereas the R49 Hδ protons are near the C$_{37}$ H5 proton (Supplementary Fig. 12a–c). Additionally, R49 makes hydrogen-bonding interactions via the guanidinium moiety with the base of G$_{28}$ as evidenced by the G$_{28}$ H8 NOE to the R49 Hη protons, thus positioning it over the remodeled TAR bulge (Fig. 5c–e). Like 7SK-SL1$^{apical}$, the remodeling of the TAR bulge also occurs via intercalation of arginine 52 into the sandwich motif. The guanidinium moiety makes hydrogen-bonding interactions with G$_{26}$ and we observe intermolecular NOEs from the aromatic protons of the U$_{23}$ cap and the A$_{22}$ base to the Hγ and Hδ protons of R52 (Fig. 5c–e and Supplementary Fig. 12a, b). Thus, the Tat:TAR interaction is stabilized by both R49 and R52, which stack over and under the cap residue U$_{23}$, respectively (Supplementary Fig. 12b).

The intervening residues K50 and K51 are also shown to be involved in stabilizing the Tat-TAR bulge interaction. For example, we observe NOEs from the H5 protons of both C$_{37}$ and U$_{38}$ to the K50 Hε proton (Fig. 5f and Supplementary Fig. 12a), as well as NOEs from the H8 protons of G$_{36}$ and G$_{34}$ to the K51 Hε proton. Below the bulge, the Tat RBD and TAR interaction continues with R53 and Q54, both of which are oriented within the major groove. The R53 sidechain protons are in close proximity to the H5 protons of C$_{39}$, U$_{40}$, and C$_{41}$ (Fig. 5g and Supplementary Fig. 12c), orienting the guanidinium moiety of R53 within hydrogen bonding distance of the U$_{40}$ backbone. Finally, the Q54 Nε protons make hydrogen-bonding interactions with the C$_{19}$-G$_{43}$ base-pair (Fig. 5g and Supplementary Fig. 12d). As the N-terminal and C-terminal residues G44-G48 and R55-Q60, respectively, do not make contact with the RNA, the Tat RBD interaction is tightly contained within a short six-amino acid (R49-Q54) stretch.

## Discussion

In summary, our lab has discovered that the cellular 7SK RNA— 7SK-SL1$^{apical}$ (this study) and 7SK-SL4[21]—is peppered with ASMs previously thought to be exclusive to retroviral TAR RNA (HIV-1, HIV-2, and BIV)[22,29,37]. The structure of the Tat RBD:7SK-SL1$^{apical}$ complex shows that the Tat RBD is able to snake through an entire helical turn of the major groove due to the multiple points of arginine intercalations that allow for numerous stacking and hydrogen-bonding interactions. Even in rare examples of protein-RNA complexes that engage the major groove, such as bovine immunodeficiency virus (BIV) Tat-TAR and HIV-1 Rev-RRE complexes, these interactions occur only within half a helical turn[22,31,38].

We also show that HIV Tat has adapted its RBD sequence to complement the orientation and placement of arginines into the four sandwiches of 7SK. Specifically, the N-terminal arginine pair, 52 and 53, interacts with the symmetrical ASM$_3$ and ASM$_4$ motifs while the C-terminal arginine pair, 56 and 57, interacts with the tandem ASM$_1$ and ASM$_2$ motifs. The same R52 responsible for switching the pseudo-ASM$_3$ configuration to ASM$_3$ for HEXIM displacement is also important for remodeling the TAR bulge into an analogous ASM with R49 making an additional stacking interaction above this induced ASM similar to that seen in the BIV Tat RBD:TAR interaction[22]. In 7SK, R49, and K50 do not form any contacts but are available for a potential TAR interaction. It is possible that these residues help facilitate the transfer of Tat RBD from 7SK to TAR, as seen in a recently modeled Tat RBD-TAR structure where R49 was placed outside the RNA[31]. Mutations in residues K50, K51, R52, R53 have been shown to have defects in HIV transactivation[17,26,27,39,40], which these studies attribute to be due to disruption of Tat-TAR interaction. However, our studies show that Tat RBD uses many of the same residues to also bind 7SK, which in fact precedes TAR binding, necessitating a reevaluating of these conclusions.

The importance of the four arginines that intercalate into 7SK is highlighted by the fact that in a virus where mutations are a frequent occurrence with Tat being one of the highly variable proteins[41], these arginines remain highly conserved (>92% conservation frequency in HIV-1 subtype B). Despite this, studies have shown that mutations to the RBDs are fairly well tolerated[26,42]. Our studies provide an explanation for this observation. We show that alanine substitutions of R57, R56, and R52 can still bind 7SK by intercalation of other ASMs. While R57 and R56 are able to remodel 7SK, the R52A mutation loses this capability. Nevertheless, it can still access pTEFb by virtue of its ability to bind 7SK via arginine intercalation into the other ASMs. An R52K mutant allows Tat to use R53 to induce the ASM$_3$ conformational switch and efficiently displace HEXIM, indicating that Tat RBD can be adaptable depending on the local environment of the residues encountering pseudo-ASM$_3$. Taken together, our studies show that having the ability to switch the pseudo-ASM$_3$ allows for active displacement of HEXIM by acquisition of a comparatively higher affinity interaction for 7SK rather than a passive, probabilistic competition between HEXIM and Tat. However, the latter option is still a viable mechanism to capture pTEFb and explains why it is not possible to completely abrogate HEXIM displacement for pTEFb capture, even upon mutations of the critical R52 insofar as the RBD is able to enter into the major groove. In keeping with this, in vitro competition experiments performed in a fully reconstituted system show that mutations of R52 and R53 to alanine results in a reduction, rather than ablation of pTEFb capture and may explain why the RBD of Tat can be replaced by the RBD of BIV despite having different arginine arrangements[20,26,42,43]. It is worth noting, however, that even if mutations were made to the entire RBD in vivo, Tat is still capable of accessing pTEFb from small complexes of pTEFb:Brd4[44]. Finally, while our work highlights the plasticity built into HEXIM displacement, our studies shows the lack of such redundancies in the TAR-Tat system, with the sole R52 intercalating into a single

TAR ASM. This may explain why mutations in R52 exhibit a significant reduction in transactivation despite active HEXIM displacement[20,27].

Importantly, our data details the mechanistic basis for the molecular mimicry by HIV at both the protein and RNA level. At the protein level, mimicry between HEXIM and Tat RBDs allows Tat to outcompete HEXIM. Tat's RBD displays a similar overall architecture to HEXIM's RBD but with one additional arginine (R52) compared to HEXIM. In fact, in many HIV strains, R52 represents the only conserved amino acid substitution compared to a lysine at that position in HEXIM (for example, KR$_{52}$KHRRR in HIV Finland strain)[13]. This limited variation between the two RBDs, which manifest as a two-fold advantage in dissociation constant for Tat, allows it to displace HEXIM in vitro. Since preforming ASM$_3$ ablates HEXIM binding, our studies suggest that HEXIM requires a pseudo-ASM$_3$ configuration to engage 7SK. Indeed, the reverse Hoogsteen A°U pair is highly conserved, suggesting that 7SK RNA has evolved to maintain a pseudo architecture in this region. By inducing the transition of this pseudo-ASM$_3$ to a canonical sandwich by virtue of R52 inter-calation, Tat is able to remodel 7SK into a configuration that is not conducive for interactions with HEXIM K151. As it is unlikely that 7SK has adapted its structure for Tat interaction, our studies suggest that HIV-1 Tat may have evolved to mimic a built-in structural switch-and-displace mechanism that allows a cellular factor(s) to displace HEXIM and extract pTEFb from 7SK for cellular transcriptional regulation. At the RNA level, HIV has evolved TAR to copy the exact secondary sequence of ASM$_3$ in 7SK-SL1$^{apical}$. While the initial tertiary structures of this bulge are different in the two RNAs, the bound structure after engagement with the same arginine residue results in identical ASM formation. This RNA structural mimicry may allow nascent TAR to dislodge 7SK from 7SK:Tat:pTEFb to form the active TAR:Tat:pTEFb ternary complex required for transcriptional elongation of the HIV genome.

While the minimal Tat and HEXIM RBDs used in this study are able to recapitulate HEXIM displacement by Tat seen both in vitro and in vivo, it is possible that other domains present in Tat, HEXIM, pTEFb, and 7SK widely influence the thermo-dynamics of binding, and may even introduce cooperativity to make the displacement more efficient. Thus, further studies with fully reconstituted 7SK snRNP complexes and full-length Tat would be required for complete assessment of the competition, although the structural elements present in our minimal system are unlikely to change under such contexts.

## Methods

**RNA sample preparation.** RNA samples used for biophysical experiments were synthesized by in vitro transcription using T7 RNA polymerase[45] with either plasmid DNA or with synthetic DNA templates containing 2'-O-methylated (Integrated DNA Technologies) containing the T7 promoter and the desired sequences (see Supplementary Table 1 for template sequences). Plasmid DNA for 7SK-SL1$^{apical}$, 7SK-SL1$^{bottom}$, and 7SK-SL1$^{top}$ contain the T7 promoter, insert, and SmaI sequence cloned in between EcoRI and BamHI restriction sites. Plasmid DNA was prepared for in vitro transcription by growing overnight a 100 mL LB starter culture of NEB 5α Competent E.coli (C29871) transformed with the plas-mid. 30 mL of the overnight starter culture is inoculated into 3 L of LB media and grown for 16 h. Cells were spun down at 4200 x g for 30 min and cell pellets were purified using Qiagen Plasmid Giga Prep (12191). Purified DNA was linearized with SmaI (R0141L) overnight on a room temperature shaker and ready for in vitro transcription the next day. To ameliorate non-physiological dimerization, the apical loop of 7SK-SL1 (nucleotides 49-59) was replaced with either a GAGA tetraloop in 7SK-SL1$^{apical/bottom}$ or stable CAGUG pentaloop in 7SK-SL1$^{top}$. Template preparation with 2'-O-methylated reverse primers, used to suppress the heterogeneity at the 3' end of the transcripts, involved combining 15μL of both forward and reverse primers at 1 mM stock solution with 47 μL of water. The mixture was heated at 95 °C for five minutes and cooled at room temperature for 30 min before assembling the in vitro transcription reaction. Samples were either unlabeled, or residue-specifically labeled with $^{13}$C/$^{15}$N- or $^2$H (Cambridge Isotope Laboratories, Inc.). After transcription, RNA samples were heat denatured and purified by using urea-denaturing polyacrylamide gels. RNA samples used for dipolar coupling measurements were first dissolved in a buffer containing 10 mM potassium phosphate, 70 mM NaCl, and 0.1 mM EDTA, pH 5.2 in D$_2$O. Following data collection without phage, the RNA samples were added to prepared Pf1 phage (Cederlane, P-200P). Pf1 phage was prepared by pelleting through centrifugation at 364637.4×g for one hour at 4 °C. The pellet was resuspended in 2 mL of the 10 mM potassium phosphate, 70 mM NaCl, and 0.1 mM EDTA, pH 5.2 in D$_2$O and pel-leted again. This solvent exchange was done a total of three times to remove residual H$_2$O before being added to concentrated RNA.

**RBD peptide preparation.** Tat RBD peptides were prepared in ABI peptide syn-thesizers by solid phase using the standard fluorenylmethoxycarbonyl (Fmoc) method while the unlabeled peptides for native Tat RBD (GISYGRKKRRQRR-RAHQ), native HEXIM RBD (GISYGRQLGKKKHRRRAHQ), Tat R52K RBD (GISYGRKKKRQRRRAHQ), Tat R52A RBD (GISYGRKKARQRRRAHQ), Tat R53A RBD (GISYGRKKRAQRRRAHQ), Tat R56A RBD (GISYGRKKRRQRAR-AHQ), Tat R57A RBD (GISYGRKKRRQRRAAHQ), and HEXIM K151R RBD (GISYGRQLGKRKHRRRAHQ), were purchased from Tufts University Core Facility. Tat peptides containing selective $^{13}$C/$^{15}$N labeled residues, underlined, (GISYGRKKRRQRRRAHQ, GISYGRKKRRQRRRAHQ, GISYGRKKRRQRR-RAHQ) were synthesized at the de Rocquigny Lab using a 0.1 mmol scale[46,47]. To incorporate selective labeled amino acids, an equimolar mixture (0.2 mmol) of labeled and non-labeled residues was added to the cartridge and the resulting peptides were purified by HPLC. The following labeled amino-acids were purchased from Cambridge Isotope Laboratories: L-Lysine-α-N-Fmoc-ε-N-T-Boc ($^{13}$C$_6$, 99%; $^{15}$N$_2$, 99%)), L-Arginine-N-Fmoc, Pbf-OH ($^{13}$C$_6$, 99%; $^{15}$N$_4$, 99%), L-Glutamine-N-Fmoc, N-γ-Trityl ($^{13}$C$_5$, 99%; $^{15}$N$_2$, 99%) and L-Alanine-N-Fmoc ($^{13}$C$_3$, 97–99%; $^{15}$N, 97–99%).

**Tat:CycT1:AFF4 preparation.** One-liter volumes of High-five cells (Thermo Fisher-B85502) were seeded at $2 \times 10^6$ cells/ml in Insect X-press media (Lonza) in 2.8 L Fernbach flasks. These were infected at an MOI of 2 with His-Tat (1–72), His-GB1- Aff (32–67) and untagged Cyclin T (1–280). The flasks were incubated at 23.5 °C with shaking at 140 rpm for 72 h. The cells were then harvested by centrifugation at 1000×g and 4 °C for 40 min in 1 L bottles. The media was decanted and the pellets were removed from the bottles with a plastic spatula and placed into Ziploc freezer bags. The pellets were stored at −80 °C until purification. For purification, the thawed cell pellet was resuspended in lysis buffer containing 50 mM sodium phosphate pH 7.4, 300 mM NaCl, 10% glycerol, 0.1% 2-mercaptoethanol and 0.1% CHAPS. Protease inhibitors and benzonase (Novagen) were added. Cells were lysed by sonication and lysate was cleared by centrifugation at 30,600×g for 1 h. The Tat/ AFF/CycT1 complex was purified by affinity chromatography with Ni-NTA resin (Qiagen). The His-tag (Tat) and His-GB1-tag (AFF) were cleaved with TEV pro-tease. The complex was run over a second Ni-NTA column then concentrated run on a Superdex200 gel filtration column (GE Healthcare Life Sciences) in buffer containing 25 mM HEPES pH 7.5, 200 mM NaCl, 1 mM DTT and 5% glycerol. The complex was flash frozen and stored at −80 °C.

**NMR data acquisition, resonance assignment and structural calculations.** For NMR experiments, the RNA samples were dissolved in a buffer containing 10 mM potassium phosphate, 70 mM NaCl, and 0.1 mM EDTA, pH 5.2. All NMR experiments were acquired by using Bruker 700 or 800 MHz instruments equipped with cryogenic probes. Spectra for observing non-exchangeable protons were col-lected at 298 K in 99.96% D$_2$O, whereas those for exchangeable protons were at 283 and 298 K in 10% D$_2$O. For NOESY experiments, mixing times were set to 200 ms. Assignments of 7SK-SL1$^{apical}$ were obtained first by transferring assignments of the 7SK-SL1$^{top}$ and 7SK-SL1$^{bottom}$ to the regions that are shared with them, and then by analyzing $^1$H–$^1$H 2D NOESY spectra of fully protonated sample. For TAR samples, NOESY datasets were recorded using unlabeled and various combinations of nucleotide-specific labeled samples[48]. Samples for 7SK-SL1$^{apical}$:Tat-RBD and TAR:Tat-RBD complex for structure determination were prepared at a 1:0.9 equivalents, respectively, to avoid any non-specific binding by the Tat RBD. Assignments for non-exchangeable $^1$H and $^{13}$C signals of 7SK-SL1$^{apical}$ and TAR free and in complex with Tat-RBD were obtained by analyzing two-dimensional $^1$H–$^1$H NOESY recorded with non-labeled samples, two-dimensional $^{13}$C-HMQC, and three-dimensional $^{13}$C-edited HMQC-NOESY spectra.

Initial structural models were generated using manually assigned restraints in CYANA[48,49]. Upper-limit distance restraints of 2.7, 3.3, and 5.0 A were employed for direct NOE cross-peaks of strong, medium and weak intensities, respectively. However, for crosspeaks pairs associated with the intra-residue H8/6 to H2' and H3', upper distance limits of 4.2 and 3.2 Å were employed for NOEs of medium and strong intensity, respectively[48]. To prevent the generation of structures with collapsed major grooves, cross-helix P–P distance restraints (with 20% weighting coefficient) were employed for A-form helical segments[48,50]. Standard torsion angle restraints were used for regions of A-helical geometry, allowing for ±50° deviations from ideality ($\alpha = -62°$, $\beta = 180°$, $\gamma = 48°$, $\delta = 83°$, $\varepsilon = -152°$, $\zeta = -73°$). Standard hydrogen bonding restraints with an approximately linear NH–N and NH–O bond distances of 1.85 ± 0.05 Å and N–N and N–O bond distances of 3.00 ± 0.05 Å, and two lower-limit restraints per base pair (G–C base pairs: G-C4 to C-C6 ≥ 8.3 Å and

G-N9 to C-H6 ≥ 10.75 Å; A–U basepairs: A-C4 to U-C6 ≥ 8.3 Å and A-N9 to U-H6 ≥ 10.75 Å) were employed in order to weakly enforce base-pair planarity (20% weighting coefficient). The twenty best CYANA models were then used for final structure calculations in AMBER[51]. The refinement was carried out in 50,000 steps, where the first 12,500 steps increased the temperature from 0 to 500 K, remained at 500 K over the next 32,500 steps, and then decreased to 0 K over the next 5000 steps. A final minimization was carried out in 8000 steps. These calculations incorporated all upper limit restraints used in CYANA but not the angle restraints. For TAR, additional tensor fitting was carried out and the above structure calculation process was repeated with the RDC restraints along with a final minimization that included 8000 steps.

**Isothermal titration calorimetry.** Binding constants for the interaction of 7SK-SL1$^{apical}$ and TAR RNAs and their mutants with the Tat and HEXIM RBDs were measured using a ITC-200 microcalorimeter (MicroCal). Briefly, 75 μM Tat or HEXIM RBD peptides were titrated into 2 μM solutions of 7SK-SL1$^{apical}$ constructs or TAR RNA in 10 mM sodium phosphate, 70 mM NaCl, 0.1 mM EDTA, pH 5.2 at 25 °C, except when analyzing the effects of salt where the NaCl concentration were varied from 50–100 mM. Experiments titrating the Tat:CycT1:AFF4 complex into 7SK-SL1$^{apical}$ or TAR was carried out by titrating 49 μM of protein complex into 2 μM solutions of RNA in 25 mM HEPES pH 7.5, 200 mM NaCl, 1 mM DTT and 5% glycerol. Titration curves were analyzed using ORIGIN (OriginLab). All thermodynamic parameters are reported with $n = 2$ experiments except for Tat:CycT1:AFF4 binding to 7SK-SL1$^{apical}$(GAGA) and 7SK-SL1$^{apical}$(Native) where $n = 3$ experiments.

**Small angle X-ray scattering.** SAXS data were obtained at SIBYLS beamline of Advanced Light Source at Lawrence Berkeley National Laboratory. Measurements were performed in buffer containing 10 mM sodium phosphate, 70 mM NaCl, 0.1 mM EDTA, pH 5.2. The background scattering was subtracted from the sample scattering to obtain the scattering intensity from the solute molecules. Data from four different concentrations (20, 30, 40, and 50 uM) were compared with scattering intensities at $q = 0$ Å$^{-1}$ [I(0)], as determined by Guinier analysis, to detect possible interparticle interactions. Data were analyzed by using ScÅtter software, and the ab initio envelope structures were reconstructed by using DAMMIF/DAMMIN software.

## Data availability

Coordinates, restraints, and structures for the final ensembles of 7SK-SL1$^{apical}$, 7SK-SL1$^{apical}$:Tat RBD and TAR: Tat RBD structures have been deposited in the Biological Magnetic Resonance Bank under 30512, 30511, and 30510 BMRB codes, and in the Protein Data Bank with PDB ID codes 6MCI, 6MCF, and 6MCE, respectively. All other data supporting the findings of this study are available from the authors upon request.

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

## Acknowledgements

We wish to thank the P50 NIH center grant (GM103297–01) and the Howard Hughes Medical Institute faculty scholar grant (55108516) for funding this project.

## Author contributions

V.V.P., C.S., S.N.K., and V.M.D.'S. conceived and designed experiments. S.N.K. cloned constructs and worked on the initial conditions for NMR and ITC experiments. N.H. and H.d.R. made the specifically labeled protein samples. V.V.P. and C.S. worked on the data analysis for 7SK-SL1. V.V.P. worked on the TAR:Tat data analysis, performed structure calculations, the final ITC experiments, and SAXS analysis. J.L.S., W.C.B., and J.L.M. cloned and prepared the cycT1:Tat:AFF4 complex; V.V.P., C.S., and V.M.D'S. wrote the manuscript.

## Additional information

**Competing interests:** The authors declare no competing interests.

