## [Peer Review File · Nature Communications]

Reviewers' comments:

Reviewer #1 (Remarks to the Author):

This manuscript is suitable for publication after the following comments are addressed.

Major

- With regards to point #12 by reviewer 1, the authors should show full 2D hsqc overlays of the two constructs in extended figure 1a and explain which parts of the structure are similar and which are modified by the mutation introduced.
- The 2-fold difference in binding affinity for 7SK to Tat and HEXIM is fairly small, especially considering the difference in Tat binding the WT 7SK and GAGA 7SK was also roughly 2-fold. Based on this, it would be important to compare the Tat and HEXIM binding affinities for the WT 7SK, and preferably the full 7SK-cycT1:AFF4 complex, rather than just the GAGA 7SK-SL1 construct. This would be important to address prior reviewer comments on biological significance.
- In reference to the mutational studies on lines 231-240, is there any NMR data on the mutated ASM constructs to determine if the mutations alter the RNA structure beyond simply disrupting the ASM motif? If there are changes in structure accompanying the mutations, this could be impacting the binding data.

Minor

- The authors should clarify what were the difficulties in measuring rDCs in this system.
- The section entitled "Tat RBD and HEXIM RBD bind with high-affinity and specificity to 7SK-SL1", only discusses Tat binding. The comparative HEXIM binding affinity should also be discussed here. Or, if it appears later in the text, this section should be renamed.
- Was the ITC data in Fig. 1c performed under the optimized salt concentrations. This would be important if 7SK is only folded properly under these conditions as stated on line 111.
- I am confused by the authors' claim: "Thus, the ability of Tat to "switch" the pseudo-ASM3 into a classical sandwich provides Tat the ability to "displace" HEXIM by changing 7SK into a conformation that precludes HEXIM binding.". Wouldn't the displacement only depend on the affinity and the Tat concentration? Why would the RNA not be able to switch back to the pre-formed context if it did prefer to bind HEXIM? Possibly the switch to the ASM is accompanied by a conformational change that induces pTEFB binding.
- Line 308: The authors state "As predicted from previous work" without citing prior work.
- Line 325 has a spelling mistake: confirmation should be conformation

Reviewer #3 (Remarks to the Author):

Structural basis of HIV-1 Tat interactions with cellular 7SK and viral TAR RNA: insights into dual molecular mimicry

Pham et al

Summary

In this NMR structure study, the authors investigate how the RNA binding domain (RBD) of HIV-1 Tat binds to two similar hairpin RNAs - cellular 7SK RNA and HIV-1 TAR RNA. They show that the Tat-binding region of 7SK RNA, SL1apical, presents a series of 4 Arginine sandwich motifs (ASMs), including a "pseudo-ASM3" due to A39 being engaged in a reverse Hoogsteen interaction with U68. They show that Tat RBD binding to SL1apical involves Arginines R57, R56, R52 and R53 being anchored in the four ASMs. Notably, Tat R52 can engage in pseudo-ASM3 and convert it into a classical ASM, which appears to be the most critical ASM in Tat RBD / 7SK-SL1apical interaction. Moreover, the authors show that Tat RBD competes with a similar HEXIM1 RBD for SL1apical binding, and they propose that Tat R52 being able to fully form ASM3 is key to HEXIM1 displacement from 7SK. This is based on their findings that (1) HEXIM RBD cannot bind SL1apical when pseudo-ASM3 is turned into a classical ASM and (2) Tat RBD R52K mutants can bind to SL1apical but not reshape ASM3. Finally, they describe the structure of a Tat RBD / TAR complex, which is based on another ASM and engage critical residues from Tat that partially overlap those also engaged in 7SK-SL1apical interaction.

Overall comments

The most interesting findings of this study are how R52 from Tat RBD can engage in and reshape pseudo-ASM3 from a 7SK-SL1apical, and how HEXIM1 RBD is unable to bind 7SK-SL1apical when ASM3 is fully formed. This provides a molecular basis for HEXIM1 displacement from 7SK by Tat, though the biological significance of these findings should be confirmed using full-length proteins and RNAs. Moreover, although the conservation of Tat R52 supports a critical role of this residue in promoting HIV transcription through displacing P-TEFb from the 7SK snRNP, the other residues from Tat RBD are not all fully conserved, and the authors should exercise more caution regarding the significance of the global Tat RBD / 7SK-SL1apical structure. The same caveat applies with TAR RNA, whose sequence also varies. The structures shown in the study thus cannot be considered as definitive for all Tat/7SK or Tat/TAR interactions, though they give interesting insights into the possible mechanisms for P-TEFb hijacking and recruitment by Tat. Regarding the form of the manuscript, some simplifications and clarifications in the main text could benefit the content. Also, the supplementary discussion is not easy to connect with the main text.

Specific comments

Regarding the answers to the points raised by Reviewer 2, we find that some were not satisfactory:

1) Reviewer 2 underlines that not all functional Tat need the four ASM-engaging Arginines described herein to successfully hijack P-TEFb from the 7SK snRNP, and asks for follow up experiments using Tat mutants. The authors consider that this is out of the scope of the paper and envision a follow-up study instead. However, we do think that using Tat RBD mutants to explore their impact on the structure of Tat RBD / 7SK-SL1apical, and the competition with HEXIM RBD for 7SK-SL1apical, would be needed for validation of the model proposed.

Later on in point 1), Reviewer 2 criticizes the use of a minimal system to study the competition between Tat and HEXIM (both being only studied as RBD peptides) for binding

to 7SK (being only studied as a partial hairpin 7SK-SL1apical with a mutated bulge), without the context of P-TEFb and TAR. The authors address this issue by adding binding analysis of Tat:CycT1: AFF4 to 7SK-SL1apical and TAR, and comparing the Kds of these interactions to that of Tat RBD with the corresponding RNAs. We wished that the authors added some data to characterize the Tat:CycT1:AFF4 complex, and show how it can recapitulate interactions from the Tat-P-TEFb complex accurately. Notably, does its structure actually allow for the RNA binding domain of CycT1 to interact with the 7SK and TAR apical loops? Some findings are troubling, such as that the Tat:CycT1:AFF4 complex binds 7SK-SL1GAGA (with a mutated loop) better than the native 7SK-SL1 (Fig.1c and lines 103-104), and that Tat RBD alone binds TAR more strongly than the Tat:CycT1:AFF4 complex (Fig.4 and lines 292-293). The physiological relevance of these experiments is questionable. Moreover, Tat:CycT1:AFF4 is only used in simple RNA binding experiments and not in competition with HEXIM as was requested.

Later on in point 1), Reviewer 2 asks how K52 Tat mutants affect the ability of Tat to shift equilibrium of large and small 7SK complexes in vivo. The authors answer that a 52 substitution has already been performed, though this was in the context of Tat RBD binding to 7SK-SL1apical only, and neither in competition with HEXIM nor in the presence of P-TEFb.

3) Reviewer 2 asks for competition experiments to be performed in both directions, and with the Tat K52 mutant, which the authors decline. They state that the outcome of the inverse competition can be predicted based on the respective Kds, however a confirmatory experiment could easily validate this, and competitions involving Tat K52 mutant could be interesting (as well as using HEXIM 151R mutants).

Additional comments

Line 70 - please clarify: what does 7SK-SL1distal correspond to? we couldn't find its description in the Materials and Methods section where the other ones are described (starting line 465). Overall, the description of the RNA domains used can be confusing / missing, and the reasons why they were specifically chosen for a given experiment are not always clear.

Line 73 - the definition of Tat RBD (G47-R57) is not consistent with line 91 (G48-R57)

Lines 99, 103-104, and later on - please confirm that there is no typo in the Kd values and standard deviations: Kd values stated as "similar" sometimes differ quite a bit, and standard deviations are sometimes almost as high as Kd values themselves. Maybe this is actually the case, but then it would be less confusing if the number of characters or decimals used was consistent. Overall, reading and comparing would be easier if all the Kd and N values were listed in a Table instead of within the main text.

Lines 262-264 - please rephrase the sentence or add a missing punctuation.

Lines 296-297 - please specify whether this data is not shown, else specify where to find it.

Line 399 - typo (missing word "by")

Line 414 - by >92% conservation, do the author means among HIV-1 subtype B? If so this should be specified as done at line 404

Line 748 - please harmonize the police size

Fig.1c - the scales are not the same in the panels, which makes comparison less straightforward (also applies to the other figures presenting ITC data). It would also be more consistent to switch panels 1 and 2 (native then GAGA), unless the legends have been switched by mistake.

Fig.3e - missing legend in this panel

In sum, both Tat:TAR and HEXIM:7SK interactions profit greatly from P-TEFb binding to apical loops of their respective stems. Distances between ASMs and these apical loops and their sequences must be conserved. Finally, the autophosphorylation of CDK9 also increases interactions between Tat, P-TEFb and TAR (and possibly 7SK?). Although not characterized in this manuscript, it is unlikely that CDK9 is present in their Tat:CycT1:AFF4 complexes (produced in insect cells). Thus, studies of ARM peptides and mutant SL1 (GAGA) are informative for ASM-binding. However, they might not recapitulate the situation in vivo, i.e. how these multiprotein complexes bind to TAR and 7SK.

Reviewers' comments

Reviewer #1 (Remarks to the Author):

This manuscript is suitable for publication after the following comments are addressed.

Major

- With regards to point #12 by reviewer 1, the authors should show full 2D hsqc overlays of the two constructs in extended figure 1a and explain which parts of the structure are similar and which are modified by the mutation introduced.

To satisfy the reviewer that changing the native loop to a GNRA tetraloop does not change the structure of the stem of 7SK-SL1^{apical}, we now show the full 2D HSQC in Extended Data Fig. 1a. The only changes are resonances attributed to the loop, which is perturbed by the presence of the GNRA.

- The 2-fold difference in binding affinity for 7SK to Tat and HEXIM is fairly small, especially considering the difference in Tat binding the WT 7SK and GAGA 7SK was also roughly 2-fold. Based on this, it would be important to compare the Tat and HEXIM binding affinities for the WT 7SK, and preferably the full 7SK-cycT1:AFF4 complex, rather than just the GAGA 7SK-SL1 construct. This would be important to address prior reviewer comments on biological significance.

We have now provided binding affinities of HEXIM to the WT 7SK-SL1^{apical} construct (pg. 9). To reduce errors, these comparative studies were done with under conditions in which the RNA and proteins were equilibrated with the same buffer preparations. We found that HEXIM binds WT 7SK-SL1^{apical} about 1.6-fold weaker than Tat, which was statistically comparable to our studies with the GAGA tetraloop construct.

While we have data for Tat:pTEFb bound to the full length 7SK, we cannot at this point do comparative experiments with full length 7SK:HEXIM:pTEFb due to complications in reconstituting the HEXIM:pTEFb complex for biophysical studies.

- In reference to the mutational studies on lines 231-240, is there any NMR data on the mutated ASM constructs to determine if the mutations alter the RNA structure beyond simply disrupting the ASM motif? If there are changes in structure accompanying the mutations, this could be impacting the binding data.

We have provided data (see Extended Data Fig. 7) to show that mutating individual ASMs do not significantly affect the integrity of the other motifs. The U₇₂A construct (ASM₂^{U72A}) did lead to broadening of the neighboring U₄₀. Nevertheless, titration of Tat RBD led to the stabilization of this motif and arginine 52 was able to intercalate in a manner exactly like the WT construct (see Extended Data Fig. 8a).

Minor

- The authors should clarify what were the difficulties in measuring rDCs in this system.

We were able to get a modest number of RDC restraints for both for the TAR:Tat complex and have now included these in the structure calculations. These RDCs were sufficient to define the groove characteristics and we have thus removed the soft phosphate restraints for this structure.

Due to the large number of bulge residues, very short stems spacing these bulges, and the overlap issues in resonances, we were not able to get enough RDC restraints for the 7SK system. For the 7SK complex, broadening of resonances was even more severe. These problems did not lead to sufficient unambiguous NOEs to refine structures. We do believe that the numerous arginine intercalations into the various ASMs and other intermolecular NOEs from the spacer residues allow for a good convergence of calculated structures.

- The section entitled “Tat RBD and HEXIM RBD bind with high-affinity and specificity to 7SK-SL1”, only discusses Tat binding. The comparative HEXIM binding affinity should also be discussed here. Or, if it appears later in the text, this section should be renamed.

We thank the reviewer for noticing the error. HEXIM is indeed introduced in another section. We have changed the title appropriately (pg. 3).

- Was the ITC data in Fig. 1c performed under the optimized salt concentrations. This would be important if 7SK is only folded properly under these conditions as stated on line 111.

All ITC were performed under optimized salt concentrations with the exception of the case where we wanted to exhibit the problems with low salt conditions. This is clarified in the methods section under “Isothermal titration calorimetry.”

- I am confused by the authors’ claim: “Thus, the ability of Tat to “switch” the pseudo-ASM3 into a classical sandwich provides Tat the ability to “displace” HEXIM by changing 7SK into a conformation that precludes HEXIM binding.” Wouldn’t the displacement only depend on the affinity and the Tat concentration? Why would the RNA not be able to switch back to the pre-formed context if it did prefer to bind HEXIM? Possibly the switch to the ASM is accompanied by a conformational change that induces pTEFB binding.

We agree with the reviewer that the displacement depends on Tat’s affinity and concentration. We only added the phrase “into a conformation that precludes HEXIM binding” because the A₃₉G, U₆₈C construct, which mimics a completely preformed ASM, does not bind HEXIM. We agree that this sentence is confusing and have now removed

the phrase “by changing 7SK into a conformation that precludes HEXIM binding” and we have discussed the relevance of this finding in more detail in the discussion section.

- Line 308: The authors state “As predicted from previous work” without citing prior work.

References added

- Line 325 has a spelling mistake: confirmation should be conformation

Corrected

Reviewer #3 (Remarks to the Author):

Structural basis of HIV-1 Tat interactions with cellular 7SK and viral TAR RNA: insights into dual molecular mimicry

Pham et al

Summary

In this NMR structure study, the authors investigate how the RNA binding domain (RBD) of HIV-1 Tat binds to two similar hairpin RNAs - cellular 7SK RNA and HIV-1 TAR RNA. They show that the Tat-binding region of 7SK RNA, SL1apical, presents a series of 4 Arginine sandwich motifs (ASMs), including a "pseudo-ASM3" due to A39 being engaged in a reverse Hoogsteen interaction with U68. They show that Tat RBD binding to SL1apical involves Arginines R57, R56, R52 and R53 being anchored in the four ASMs. Notably, Tat R52 can engage in pseudo-ASM3 and convert it into a classical ASM, which appears to be the most critical ASM in Tat RBD / 7SK-SL1apical interaction. Moreover, the authors show that Tat RBD competes with a similar HEXIM1 RBD for SL1apical binding, and they propose that Tat R52 being able to fully form ASM3 is key to HEXIM1 displacement from 7SK. This is based on their findings that (1) HEXIM RBD cannot bind SL1apical when pseudo-ASM3 is turned into a classical ASM and (2) Tat RBD R52K mutants can bind to SL1apical but not reshape ASM3. Finally, they describe the structure of a Tat RBD / TAR complex, which is based on another ASM and engage critical residues from Tat that partially overlap those also engaged in 7SK-SL1apical interaction.

Overall comments

The most interesting findings of this study are how R52 from Tat RBD can engage in and reshape pseudo-ASM3 from a 7SK-SL1apical, and how HEXIM1 RBD is unable to bind 7SK-SL1apical when ASM3 is fully formed. This provides a molecular basis for HEXIM1 displacement from 7SK by Tat, though the biological significance of these findings should be confirmed using full-length proteins and RNAs. Moreover, although the conservation of Tat R52 supports a critical role of this residue in promoting HIV transcription through displacing P-TEFb from the 7SK snRNP, the other residues from

Tat RBD are not all fully conserved, and the authors should exercise more caution regarding the significance of the global Tat RBD / 7SK-SL1^{apical} structure. The same caveat applies with TAR RNA, whose sequence also varies. The structures shown in the study thus cannot be considered as definitive for all Tat/7SK or Tat/TAR interactions, though they give interesting insights into the possible mechanisms for P-TEFb hijacking and recruitment by Tat. Regarding the form of the manuscript, some simplifications and clarifications in the main text could benefit the content. Also, the supplementary discussion is not easy to connect with the main text.

We agree with the reviewer that a more robust discussion is needed to highlight the intricacies of the Tat transactivation system. We have attempted to provide more context for our findings in the discussion section.

We have also added a phrase in the paper to clarify what the content of the supplementary discussion contains.

Specific comments

Regarding the answers to the points raised by Reviewer 2, we find that some were not satisfactory:

1) Reviewer 2 underlines that not all functional Tat need the four ASM-engaging Arginines described herein to successfully hijack P-TEFb from the 7SK snRNP, and asks for follow up experiments using Tat mutants. The authors consider that this is out of the scope of the paper and envision a follow-up study instead. However, we do think that using Tat RBD mutants to explore their impact on the structure of Tat RBD / 7SK-SL1^{apical}, and the competition with HEXIM RBD for 7SK-SL1^{apical}, would be needed for validation of the model proposed.

As requested, we have now performed experiments (NMR assignments and competition experiments with HEXIM RBD) with Tat RBD mutations at residues R57, R56, R52, and R53, which intercalate into the four ASMs of 7SK-SL1^{apical}. Of all the constructs, only R53A had an uncharacteristic behavior wherein even the N-terminal and C-terminal residues that do not contact the RNA and give sharp line widths in the wild-type construct exhibited severe line broadening; hence this data was difficult to interpret and is not further analyzed in the manuscript.

In brief, we confirm that: a) R57A, R56A, R52A, and R52K mutant constructs are able to bind 7SK-SL1^{apical} by intercalation of the remaining arginines. Importantly, the local interactions of individual ASMs do not differ significantly from what is seen in the wild-type RBD, indicating that these mutations do not affect the register of binding; b) competition experiments of these constructs into the 7SK-SL1^{apical}:HEXIM complex led to varying degrees of displacement efficiencies. The relevance of this is discussed.

Later on in point 1), Reviewer 2 criticizes the use of a minimal system to study the competition between Tat and HEXIM (both being only studied as RBD peptides) for binding to 7SK (being only studied as a partial hairpin 7SK-SL1^{apical} with a mutated bulge), without the context of P-TEFb and TAR. The authors address this issue by adding

binding analysis of Tat:CycT1: AFF4 to 7SK-SL1^{apical} and TAR, and comparing the K_ds of these interactions to that of Tat RBD with the corresponding RNAs. We wished that the authors added some data to characterize the Tat:CycT1:AFF4 complex, and show how it can recapitulate interactions from the Tat-P-TEFb complex accurately. Notably, does its structure actually allow for the RNA binding domain of CycT1 to interact with the 7SK and TAR apical loops?

The native 7SK construct at NMR concentrations has significant levels of duplex formation via the loop. This complicates analysis. However, we have now included a titration of the Tat:CycT1:AFF4 with TAR RNA and show that in addition to the engagement of the ASM domain, the TAR loop is also bound and experiences shifts in residues previously shown to engage pTEFb¹ (Extended Data Fig. 10a).

Some findings are troubling, such as that the Tat:CycT1:AFF4 complex binds 7SK-SL1GAGA (with a mutated loop) better than the native 7SK-SL1 (Fig.1c and lines 103-104), and that Tat RBD alone binds TAR more strongly than the Tat:CycT1:AFF4 complex (Fig.4 and lines 292-293). The physiological relevance of these experiments is questionable.

In the previous iteration of the paper, we stated that the binding affinities of the Tat:CycT1:AFF4 complex to both WT and GAGA 7SK-SL1^{apical} constructs were comparable. We have now included an additional dataset to increase N-values to 3 and have updated our values in the paper. T-tests on these values confirm that the differences are not statistically significant. Similarly, Tat RBD alone and the Tat:CycT1:AFF4 complex differences in binding affinities are also not statistically significant using a t-test.

However, we were confused by the logic that Tat:CycT1:AFF4 affinity to TAR must be tighter than the Tat RBD simply because CycT1 is going to interact with the native loop. Binding affinities arise from enthalpic/entropic compensations. Thus, while the CycT1 will of course contribute additional enthalpic interactions with the loop, it is also possible that the free loop will lose its degrees of freedom upon binding and therefore the whole system may reduce in entropy. Thus, one cannot predict the affinities of complex interactions. The same argument can also be made for the 7SK-SL1^{apical} GAGA versus WT binding to Tat:CycT1:AFF4. In fact, this is evidence in our ITC analysis where the enthalpic contributions of the Tat:CycT1:AFF4 to the Native loop is higher than that of the GAGA construct, suggesting that this interaction does occur. In both of these cases, the observed differences in K_d are not of a concern because they are within errors of each other.

One could argue that the same logic should preclude us from using the minimal RBD domains to understand the mechanistic basis of HEXIM displacement since the presence of pTEFb and full length 7SK could hypothetically lead to tighter HEXIM affinities compared to Tat. However, our studies are justified because it has been shown both *in vitro* and *in vivo* that it is Tat that is able to displace HEXIM from 7SK via its RBD domain and not the other way around²⁻⁵.

Moreover, Tat:CycT1:AFF4 is only used in simple RNA binding experiments and not in competition with HEXIM as was requested.

We are assuming that the reviewer wants us to show that the Tat:CycT1:AFF4 complex can induce the conformation change from a pseudo to a classical ASM configuration upon titration into a HEXIM:7SK-SL1^{apical} complex. However, this is not a relevant experimental setup because Tat forms this complex only after the displacement of HEXIM. To perform the relevant competition experiment with HEXIM, one would have to fully reconstitute the 7SK snRNP complex with HEXIM:pTEFb and titrate in Tat. This experiment would not be feasible by NMR because of the sizes of the molecules involved. Nevertheless, this competition experiment has been performed by other labs both *in vivo* and *in vitro* with such a minimal system and indeed Tat is able to displace HEXIM²⁻⁵. The conclusions from these studies are that the Tat pTEFb-binding domain and the RBD both contribute to HEXIM displacement. The goal of this paper is to get the mechanistic insight of Tat RBD's contribution to HEXIM displacement, especially since Muniz et al. showed that Tat and HEXIM bind 7SK in a mutually exclusive manner⁵.

Later on in point 1), Reviewer 2 asks how K52 Tat mutants affect the ability of Tat to shift equilibrium of large and small 7SK complexes *in vivo*. The authors answer that a 52 substitution has already been performed, though this was in the context of Tat RBD binding to 7SK-SL1^{apical} only, and neither in competition with HEXIM nor in the presence of P-TEFb.

We now include a complete analysis of the R52K mutation both in its mode of binding to 7SK-SL1^{apical} and in competition with HEXIM RBD (Extended Data Fig. 8f and 9). This mutant is able to bind 7SK-SL1^{apical} in the proper orientation and uses R53 to switch from the pseudo to a classical ASM. This interesting findings and its implications is discussed.

The experiments in the presence of pTEFb:HEXIM competition are not feasible via NMR. To make comparisons with a previously performed *in vitro* analysis on a minimal system³, we made an R52A mutation, which although was unable to switch the pseudo ASM, and equilibrated with native HEXIM. This construct has been shown *in vitro* to slow down the release of pTEFb from 7SK snRNP³. Our data explains why this mutation does not completely abrogate HEXIM displacement. We see that both R52A are able to allow for intercalations of other arginines into the ASMs. By making these mutations, we are simply turning Tat RBD into a HEXIM-like RBD and it is going to be capable of binding to 7SK-SL1^{apical} and acquiring pTEFb.

3) Reviewer 2 asks for competition experiments to be performed in both directions, and with the Tat K52 mutant, which the authors decline. They state that the outcome of the inverse competition can be predicted based on the respective Kds, however a confirmatory experiment could easily validate this, and competitions involving Tat K52 mutant could be interesting (as well as using HEXIM 151R mutants).

We have done the reverse titration and, as predicted, titration of HEXIM into the 7SK-SL1^{apical}. Tat complex was not able to displace Tat. We have provided an extra panel in Fig. 3e to demonstrate this. As explained in the above comments, we have completed competition experiments with Tat R52K (Extended Data Fig. 9). Finally, we have also performed experiments with the HEXIM151R mutant and show that it is able to: 1) switch the pseudo configuration into a classical ASM, and 2) this construct performs better than native HEXIM in a competition assay (Extended Data Fig. 8f and 9).

Additional comments

Line 70 - please clarify: what does 7SK-SL1^{distal} correspond to? We couldn't find its description in the Materials and Methods section where the other ones are described (starting line 465). Overall, the description of the RNA domains used can be confusing / missing, and the reasons why they were specifically chosen for a given experiment are not always clear.

Done. We apologize for the typo. There is no such construct as 7SK-SL1^{distal}.

Line 73 - the definition of Tat RBD (G47-R57) is not consistent with line 91 (G48-R57)

Done. We have now corrected G47 and G48 to reflect that Tat RBD is from G48-R57.

Lines 99, 103-104, and later on - please confirm that there is no typo in the K_d values and standard deviations: K_d values stated as "similar" sometimes differ quite a bit, and standard deviations are sometimes almost as high as K_d values themselves. Maybe this is actually the case, but then it would be less confusing if the number of characters or decimals used was consistent. Overall, reading and comparing would be easier if all the K_d and N values were listed in a Table instead of within the main text.

We have now changed the K_d values to reflect a consistent decimal value and have also provided a table (Table 2) in addition to the description in the text. We have also removed the word "similar" from the text. We agree that the wordings were not clear and simply meant that the bindings are in the lower nanomolar range for both constructs.

Lines 262-264 - please rephrase the sentence or add a missing punctuation.

Done. We have added a comma for clarity.

Lines 296-297 - please specify whether this data is not shown, else specify where to find it.

Done. We have added, "data not shown".

Line 399 - typo (missing word "by")

Done.

Line 414 - by >92% conservation, do the author means among HIV-1 subtype B? If so this should be specified as done at line 404

We have clarified that the >92% conservation is among HIV-1 subtype B.

Line 748 - please harmonize the police size

We have changed the font size to match the rest of the text in the section.

Fig.1c - the scales are not the same in the panels, which makes comparison less straightforward (also applies to the other figures presenting ITC data). It would also be more consistent to switch panels 1 and 2 (native then GAGA), unless the legends have been switched by mistake.

We have switched panels 1 and 2 to be more consistent with panels 3 and 4.

Fig.3e - missing legend in this panel

We have now added a label to clarify this figure.

In sum, both Tat:TAR and HEXIM:7SK interactions profit greatly from P-TEFb binding to apical loops of their respective stems. Distances between ASMs and these apical loops and their sequences must be conserved. Finally, the autophosphorylation of CDK9 also increases interactions between Tat, P-TEFb and TAR (and possibly 7SK?). Although not characterized in this manuscript, it is unlikely that CDK9 is present in their Tat:CycT1:AFF4 complexes (produced in insect cells). Thus, studies of ARM peptides and mutant SL1 (GAGA) are informative for ASM-binding. However, they might not recapitulate the situation in vivo, i.e. how these multiprotein complexes bind to TAR and 7SK.

References

- 1 Schulze-Gahmen, U. *et al.* Insights into HIV-1 proviral transcription from integrative structure and dynamics of the Tat:AFF4:P-TEFb:TAR complex. *Elife* **5**, doi:10.7554/eLife.15910 (2016).
- 2 Barboric, M. *et al.* Tat competes with HEXIM1 to increase the active pool of P-TEFb for HIV-1 transcription. *Nucleic Acids Res* **35**, 2003-2012, doi:10.1093/nar/gkm063 (2007).
- 3 Krueger, B. J., Varzavand, K., Cooper, J. J. & Price, D. H. The mechanism of release of P-TEFb and HEXIM1 from the 7SK snRNP by viral and cellular activators includes a conformational change in 7SK. *PLoS One* **5**, e12335, doi:10.1371/journal.pone.0012335 (2010).
- 4 Sedore, S. C. *et al.* Manipulation of P-TEFb control machinery by HIV: recruitment of P-TEFb from the large form by Tat and binding of HEXIM1 to TAR. *Nucleic Acids Res* **35**, 4347-4358, doi:10.1093/nar/gkm443 (2007).

- 5 Muniz, L., Egloff, S., Ughy, B., Jady, B. E. & Kiss, T. Controlling cellular P-TEFb activity by the HIV-1 transcriptional transactivator Tat. *PLoS Pathog* **6**, e1001152, doi:10.1371/journal.ppat.1001152 (2010).

REVIEWERS' COMMENTS:

Reviewer #1 (Remarks to the Author):

The revised manuscript is much improved and is acceptable for publication.

Reviewer #3 (Remarks to the Author):

Despite some additional data, our main concerns had not been addressed and it is not clear that NMR alone can add substantially to our understanding of how Tat:P-TEFb interact with TAR and compete with HEXIM1:P-TEFb for 7SK snRNA. It is after all cooperative interactions between respective complexes that play critical roles in these RNA:protein interactions.

RESPONSE TO REVIEWER COMMENTS

Reviewer #1 (Remarks to the Author):

The revised manuscript is much improved and is acceptable for publication.

Reviewer #3 (Remarks to the Author):

Despite some additional data, our main concerns had not been addressed and it is not clear that NMR alone can add substantially to our understanding of how Tat:P-TEFb interact with TAR and compete with HEXIM1:P-TEFb for 7SK snRNA. It is after all cooperative interactions between respective complexes that play critical roles in these RNA:protein interactions.

We have now added a paragraph at the end of the discussion to address the importance of looking at full length constructs for further validation of our structural findings.